# Advances in Pyranopyrazole Scaffolds’ Syntheses Using Sustainable Catalysts—A Review

**DOI:** 10.3390/molecules26113270

**Published:** 2021-05-28

**Authors:** Ravi Kumar Ganta, Nagaraju Kerru, Suresh Maddila, Sreekantha B. Jonnalagadda

**Affiliations:** 1Department of Chemistry, GITAM Institute of Sciences, GITAM University, Visakhapatnam 530045, India; rganta@gitam.edu (R.K.G.); sureshmskt@gmail.com (S.M.); 2Department of Chemistry, GITAM School of Science, Bengaluru Campus, GITAM University, Karnataka 561203, India; nagarajukerru@gmail.com; 3School of Chemistry & Physics, Westville Campus, University of KwaZulu-Natal, Chiltern Hills, Durban 4000, South Africa

**Keywords:** multicomponent reactions, heterogeneous catalysts, pyranopyrazoles, one-pot process, green protocols

## Abstract

Heterogeneous catalysis plays a crucial role in many chemical processes, including advanced organic preparations and the design and synthesis of new organic moieties. Efficient and sustainable catalysts are vital to ecological and fiscal viability. This is why green multicomponent reaction (MCR) approaches have gained prominence. Owing to a broad range of pharmacological applications, pyranopyrazole syntheses (through the one-pot strategy, employing sustainable heterogeneous catalysts) have received immense attention. This review aimed to emphasise recent developments in synthesising nitrogen-based fused heterocyclic ring frameworks, exploring diverse recyclable catalysts. The article focused on the synthetic protocols used between 2010 and 2020 using different single, bi- and tri-metallic materials and nanocomposites as reusable catalysts. This review designated the catalysts’ efficacy and activity in product yields, reaction time, and reusability. The MCR green methodologies (in conjunction with recyclable catalyst materials) proved eco-friendly and ideal, with a broad scope that could feasibly lead to advancements in organic synthesis.

## 1. Introduction

The strategy of using sustainable green methodologies to synthesise various heterocycles utilising heterogeneous catalysts is a continuously expanding area of interest for researchers and industries [1,2,3,4]. Over the years, synthetic organic chemists explored many heterogeneous catalysts due to their distinctive characteristics, such as thermal stability, durable absorption ability, selectivity, low toxicity, easy modification, tunable surface and textual properties [5,6,7]. Additionally, easy recovery and recyclability (compared to homogeneous catalysts) sustained interest in them [8,9,10]. The tunable strong acid and base properties and catalytic activity of heterogeneous materials played a crucial role in organic synthesis [10,11,12]. Generally, it is challenging to design and develop eco-friendly, cost-effective and stable composites that show good activity. Diverse conceptualisations have proven helpful in interpreting activity associated with different heterogeneous catalysts. In supported catalyst materials, the active phase experience phase-support associations [11]. The surface free dynamics of the active phase, with catalytic support materials and the interfacial free energy between the two constituents, help such interactions. Transition metal oxides possess considerably low surface free energies, similar to support materials like alumina, titania, zirconia, and silica [9,10,11,12]. The active transition metal oxide develops a monolayer through the wetting of the support surface. As transition metals possess large surface free energy, tiny metal particles lead to aggregation, diminishing their surface area [6]. Hence, stabilising the nano-size metal particles necessitates their deposition on the support surface, enabling convenient metal-support synergies. The physical characteristics and morphology are primarily influenced by these interactions with depreciating particle size. Therefore, the nature of the support material significantly controls the catalytic behaviour of metal particles.

On the other hand, multicomponent reactions (MCRs) have steadily gained significance in synthetic organic chemistry [13,14,15,16,17,18,19]. MCRs offer many benefits—e.g., greater yields, high atom efficiency, reduced reaction time, high convergence, minimisation of purification requirements and waste generation—compared to multiple-step syntheses [20,21,22,23,24,25]. Typically, multi-step procedures require the purification of reaction products in each step and demand appropriate solvents and reagents. Hence, such processes generate more chemical waste, affecting the environment [26,27,28]. The one-pot approach, however, uses effective reusable catalysts, green solvents and eco-friendly methods [29,30,31,32]. Using efficient heterogeneous catalyst materials is a progressive approach to generate desired products with high selectivity and atom economy [33,34]. Researchers have made vast progress in designing and developing libraries of bioactive molecules comprising different heterocyclic structures utilising the MCR approach over the past two decades [35,36]. Through the choice of reactants, the MCR approach can actively introduce a chromophore into a scaffold (i.e., scaffold approach) or facilitate a chromogenic event (i.e., chromophore approach) (Figure 1). These developments encouraged the generation of high-functional and structurally diverse molecules. Many MCRs are achievable utilising flexible and easily available starting materials to permit target compounds, and are thus proving a versatile tool in medicinal, pharmaceutical, agrochemical, computational and material sciences [37,38,39,40].

Heterocycles are important components of many natural materials and are extremely valuable in organic and medicinal chemistry [41,42,43,44,45]. Over the years, biological and therapeutic arenas recognised the importance of heterocyclic scaffolds [46,47,48]. Furthermore, combinatorial chemistry also accelerated the creation of chemical entities with elite structural units. Among the heterocyclic entities, pyranopyrazole moieties have demonstrated remarkable biochemical behaviours and activities which provide a versatile skeleton for drug innovation. Hence, many nitrogen-based, fused structures have been incorporated as building blocks of various pharmacological potent scaffolds [49,50,51]. Pyranopyrazoles are known for their anti-inflammatory, analgesic, antidiabetic, antimicrobial, cholinesterase-inhibiting, antibacterial and anticancer activities, as well as for their efficacy in treating Alzheimer’s disease [52,53,54,55,56,57]. Because of this, several cost-effective synthetic protocols for synthesising pyranopyrazole derivatives—utilising less expensive substrates, reusable catalysts, and eco-friendly solvents—have been developed. This review set out to provide insight into the synthesis procedures for different pyranopyrazole analogues via the MCR approach, using various recyclable materials and nanocomposites as catalysts. This study also emphasised the catalyst’s efficacy towards the precursors, conversion, product selectivity and reaction time.

Pyranopyrazole is a fused heterocyclic framework comprising pyran and pyrazole moieties. Pyranopyrazoles exhibit four isomeric forms, namely: pyrano[3,2-*c*]-pyrazoles **1**, pyrano[3,4-*c*]-pyrazoles **2**, pyrano[4,3-*c*]-pyrazoles **3** and pyrano[2,3-*c*]-pyrazoles **4** (Figure 2). Among the four isomers, 4*H*-pyrano[2,3-*c*]-pyrazole is the most privileged structure, due to its versatile biological profile, as first reported by H.H. Otto in 1973 using sodium methylate as a catalyst [58].

This review focused on the protocols used for synthesis of different 4*H*-pyrano[2,3-c]-pyrazole analogues using various metal-based materials and composites, grouping methodologies as (i) single metal-containing, (ii) two-metal containing, (iii) three-metal containing and (iv) with miscellaneous materials and composites as recyclable catalysts.

## 2. Single Metal-Containing Catalysts

Several single metal-based as catalysts have been developed as catalysts for carbon-carbon or carbon-heteroatom-making conversions. The cost-effective and eco-friendly supports associated with these metals further enhance their candidacy as one of the most-desired catalysts for medicinal and chemical productions. The appeal of transitioning to metal-catalysed reactions is due to the robust activity and selectivity achievable. Another advantage of MCRs is their potential to utilise mono metal oxide catalysts and composites entirely.These mono metal oxide catalysts are pigeon-holed with the higher surface-to-volume percentages of the active metal species, improving the material’s efficacy. Moreover, highly distributed metal oxide particles are presumed to perform as additional energetic and choosy catalysts more than most materials. Additionally, owing to their extraordinary surface energies, minor particles tend to cumulate easily.

Babaie et al. [59] reported a facile, efficient synthesis of dihydropyrano[2,3-*c*]-pyrazole derivatives (**5**) from the reaction of different aldehydes—(**1**) substituted hydrazine (**2**), malononitrile (**3**), and 3-oxoproponoate (**4**)—catalysed by nanosized magnesium oxide (MgO). A total of 50 mg of nanocatalyst performed well in an aqueous medium for 20 min reaction time at room temperature (RT) and afforded 88–97% yields of the desired products (Scheme 1). Compared with the commercially available MgO, the as-synthesised MgO nanocatalyst showed superior catalytic activity with yield and reaction time. The authors reported excellent yields with the electron-donating and -withdrawing substituents on the aldehydes.

Hasaninejada et al. [60] synthesised a novel series of pyrano-[2,3-*c*]-pyrazole analogues (**9**) in the presence of SiO_2_ supported *n*-propyl-4-aza-1-azoniabicyclo-[2.2.2]-octane chloride (SB-DABCO). The multicomponent reaction was conducted using various substituted aldehydes (**6**), different active methylene compounds (**7**) and 3-methyl-pyrazolone (**8**) in ethanol as solvent medium (Scheme 2). Excellent yields (90–98%) were obtained using 6 mol% of SB-DABCO catalyst at room temperature for 35 min, due to the relative strength of basic sites and active surface area (~160 m^2^ kg^−^^1)^. Their protocol offered several benefits including simple handling, greater stability, and an easy workup, and was recyclable up to five runs without loss of catalytic activity. This protocol offered high yields with all three active methylene compounds (malononitrile, methyl cyanoacetate and ethyl cyanoacetate).

Shaterian and co-workers [61] developed titanium dioxide as a nanocatalyst for the generation of dihydropyrano-[2,3-*c*]-pyrazoles (**13**) through a one-step process. The condensation was carried under solvent-free conditions at RT by reacting the substituted aldehydes (**10**), malononitrile (**3**), hydrazine hydrate (**11**) and ethyl acetoacetate (**12**). Additionally, 0.25 mmol of nano-titania showed excellent product yields (81–96%) (Scheme 3). The proposed reaction mechanism involved a Knoevenagel condensation of malononitrile and aldehyde form 2-benzylidenemalononitrile as intermediate. A Michael addition, followed by intramolecular cyclisation of pyrazolone with the intermediate, delivered the final product. The nanocatalyst material exhibited similar activity up to the fifth cycle.

Azarifar et al. [62] developed the nanocatalysts—titania-loaded Pressler-type hetero-polyacid (nTiO_2_/H_14_[NaP_5_W_30_O_110_])—as a catalyst by sol-gel approach. The catalyst’s structure was characterised and confirmed by BET and TEM analysis. TEM analysis confirmed nanoparticles with an average size (15–50 nm) and BET-specific surface area (175 m^2^g^−1^). The nano TiO_2_/H_14_[NaP_5_W_30_O_110_] material quickly catalysed the three-component reaction of malononitrile (**3**), aromatic aldehydes (**14**), 3-methyl-pyrazolinone (**8**) and 25 mg catalyst, along with ultrasound irradiation at 40 °C in ethanol solvent medium (Scheme 4). The target dihydropyrane[2,3-*c*]pyrazoles (**15**) were synthesised by good-to-excellent yields due to the higher surface area of catalyst particles.

In 2014, Paul and his co-workers [63] reported the novel synthesis of pyrano[2,3-*c*] pyrazole derivatives (**20**) and spiro-pyranopyrazoles (**21**) using uncapped SnO_2_ quantum dots (QDs). The catalyst was synthesised by solvothermal technique, and the resultant catalyst was confirmed by XRD, TEM and SEM spectroscopy analysis. The XRD, SEM, and TEM analyses revealed that the SnO_2_ QDs (with an average size of 3.9 nm) had an equally uniform nanoflower spherical shape size (with 100 nm), corresponding to the lattice plane (110). The four-component condensation reaction involved malononitrile (**3**), substituted hydrazine (**16**), dialkyl acetylenedicarboxylates (**17**) and aldehydes (**18**) or substituted isatins (**19**) in the aqueous medium. The high Lewis acidic character and surface area of the Sn^+4^ catalyst (8 mol%) performed well and provided the anticipated target molecules at room temperature for 2.5 h reaction time with excellent yields (89–98%) (Scheme 5). The catalyst material was environmentally friendly and stable for up to six cycles with sustained selectivity and activity. Both aldehyde (89–98%) and isatin (91–95%) substrates were executed well and offered significant product yields. Low catalyst-loading, eco-friendly conditions, a broad substrate scope, a simple workup and the use of a water medium were the advantages of this procedure.

Borhade et al. [64] designed a ZnS nanoparticle catalyst by hydrothermal method for synthesising pyrano[2,3-*c*]-pyrazoles (**23**). P-XRD, TEM, SEM and BET microscopic spectroscopic analysis validated the structure of the subsequent ZnS nanoparticles. The prepared ZnS nanoparticle’s P-XRD pattern displayed a single-phase hexagonal arrangement with crystallite size (20 nm) and was confirmed with TEM and SEM analysis. Furthermore, the BET spectrum revealed that the nanoparticle specific surface area was (84.71 m^2^.g^−1^), the pore volume was (0.0865 cc g^−1^) and diameter was (31.11 Å). The four-component reaction between aromatic aldehydes (**22**), hydrazine hydrate (**11**), ethyl acetoacetate (**12**), and malononitrile (**3**) under solvent-free and grinding conditions was efficiently accelerated by ZnS nanoparticles, affording excellent yields (87–97%) in 12 min (Scheme 6). The catalyst material could be simply separated by filtration and reused for up to five successive runs without loss of its catalytic activity. The attractive features of this procedure were the low catalyst requirement and recyclability and the mild, eco-friendly solvent-free reaction conditions.

Irvani et al. [65] developed novel tin sulfide nanoparticles on activated carbon [SnS-NPs@AC] and explored them as a catalyst preparing pyrano-[2,3-*c*]-pyrazoles (**25**). The composite material and solvent influenced the optimised conditions. The spectroscopic analysis (P-XRD, TEM and SEM. TEM and SEM) of the prepared [SnS-NPs@AC] specified that the nanoparticles had a homogeneously spherical morphology and were in the range of (40–90 nm) and (30–70 nm), respectively, which was in agreement with the PXRD (64 nm) outcomes. The multicomponent reaction of 3-methyl-pyrazolinone (**8**), various aldehydes (**24**), malononitrile (**3**) and catalyst in ethanol solvent led to the target compounds at 80 °C for 25 min reaction time with excellent yields (85–91%) (Scheme 7). The nanocatalyst performed eight cycles with sustained activity. The mechanistic scheme comprised Knoevenagel condensation, Michael-type addition, and cyclisation to yield the target products.

Zainali et al. [66] published the highly efficient, one-pot, rapid synthesis of pyrano[2,3-*c*]-pyrazole derivatives (**28**) in the presence of amino-functionalised SBA-15 (SBA-Pr-NH_2_) catalyst. The multicomponent reaction was performed using arylmethlidenemalononitrile (**26**), phenylhydrazine (**27**) and ethyl acetoacetate (**12**) in ethanol at RT for 8 min with excellent yields (80–95%) (Scheme 8). The possible mechanism described suggested that the intermediate formed through Knoevenagel condensation of phenylhydrazine with ethyl acetoacetate, then underwent Michael addition with arylmethlidene malononitrile to afford the target molecules.

Beerappa and his colleagues [67] investigated a novel synthesis of a library of pyranopyrazole scaffolds (**32**) using *N*-methyl-morpholine *N*-oxide-doped silver oxide (NMO-Ag_2_O) catalyst. The condensation reaction was progressed using benzyl halide (**29**), active methylene compound (**30**), dialkylacetylenedicarboxylate (**31**) and hydrazine hydrate (**11**) in ethanol solvent medium under reflux condition for 1 h reaction time (Scheme 9). The reported mechanism involved the initial benzyl halide being converted into benzaldehyde (**33**) in the presence of a catalyst. It proceeded with Knoevenagel condensation (**34**). The obtained pyrazolone (**35**) involved Michael’s addition (**36**) and was followed by cyclisation (**37**) to afford target compounds.

Patel et al. [68] completed the preparation of pyrano[2,3-*c*]-pyrazoles (**39**) by applying a recyclable nano-SiO_2_ catalyst. The nano-silica catalyst was synthesised from the agriculture waste of wheat straw via the sol-gel process. The results indicated that the prepared catalyst comprised a spherical shape with uniform distribution and crystallite size range of 100–200 nm. The BET analysis showed the surface area (215.6 m^2^ g^−1^), pore volume (0.269 cm^3^ g^−1^) and pore diameter (7.1 nm) for the catalyst. The multicomponent reaction involved hydrazine hydrate (**11**), malononitrile (**3**), aromatic aldehydes (**38**) and ethyl acetoacetate (**12**) in water. Additionally, 10 mol% of nanocatalyst showed the best performance and offered 87–94% yields for 40 min reaction time at 80 °C (Scheme 10). The agriculture waste catalyst was fully stable for up to five runs without significant loss of activity.

Salehi et al. [69] reported one-pot and efficient synthesis of two series of pyrano[2,3-*c*]-pyrazoles (**43**) and spiro-pyrano[2,3-*c*]-pyrazoles (**44**) using a nano SiO_2_-supported 1,4-diazabicyclo-[2.2.2]-octane (nano SiO_2_/DABCO) catalyst. The SEM-EDX showed that the synthesised SiO_2_/DABCO nanomaterial contained C, Cl, N, O and Si elements. The catalyst activity was investigated on the multicomponent reaction of the beta-keto ester (**12**), substituted hydrazine (**40**), malononitrile (**3**) and aldehyde (**41**) or isatin (**42**) in the absence of solvent at RT via the grinding method (Scheme 11). Recycling investigation showed no significant reduction in the catalytic activity even after six cycles, giving excellent yields (80–95%). The reaction progressed in sequential steps, involving Knoevenagel condensation and Michael addition, followed by intramolecular cyclisation.

Ghasemzadeh et al. [70] reported a novel synthesis of spiro-pyrano[2,3-*c*]-pyrazole derivatives (**49**) using Fe_3_O_4_@L-arginine as a heterogeneous catalyst via one-pot manner. The structure of the prepared nanocatalyst was analysed by P-XRD, FT-IR, SEM and VSM spectroscopy methods. SEM investigation indicated that the catalyst particles were made in circular shapes with an average size of (10–15 nm). TEM study confirmed that the morphology and nanoparticle average size (10–20 nm) was established by P-XRD analysis. The four-component condensation carried out between the substituted hydrazines (**45**), *β*-keto esters (**46**), various functional groups substituted isatin (**47**) and active methylene compound (**48**) under solvent-free conditions. The use of 8 mol% of nanocatalyst offered excellent yields (86–97%) of the target products at room temperature after a 60 min reaction time (Scheme 12). The nanocatalyst was recycled up to five times without significant loss of its catalytic efficiency. Active methylene compounds, malononitrile and ethyl cyanoacetate endured well, giving high yields.

Mianai and co-workers [71] reported the synthesis of novel pyrano-[2,3-*c*]-pyrazole frameworks (**52**) in the presence of cobalt nanoparticles (CoNPs) as heterogeneous catalyst. The cobalt nanoparticles were prepared from the aqueous extract of *zingiber*. SEM analysis showed that the morphology and homogeneous nanoparticles had an average size (20–50 nm). The EDS investigation affirmed the existence of elements in the prepared catalyst. The condensation reaction was carried out between the hydrazine hydrate (**11**), malononitrile (**3**), diethyl acetylenedicarboxylate (**50**) and substituted aldehydes (**51**) in an aqueous ethanol medium. The use of 0.005 g of nanocatalyst performed well and afforded excellent yields (83–97%) of desired products after a 1 h reaction time at room temperature (Scheme 13). Mild reaction conditions, easy workup, high yields, and reusability were the essential advantages of this approach.

Tabassum et al. [72] synthesised a novel series of pyrano[2,3-*c*]-pyrazoles (**55**), applying ZnO@PEG as a nanocatalyst. Various aromatic aldehydes having electron-donating/electron-withdrawing substituents (**54**) were reacted with ethyl acetoacetate (**12**), hydrazine hydrate (**11**), and 4-nitro phenyl acetonitrile (**53**) in ethanol under ultrasound-assisted conditions, affording excellent 87–97% yields in 15 min (Scheme 14). Furthermore, it was noteworthy that the nanocatalyst was recycled and reused for more than five runs with only a minor loss in its activity.

Abbasabadi et al. [73] described the highly efficient sulfonic acid-mobilised Fe_3_O_4_-graphene oxide magnetic catalyst (Fe_3_O_4_@GO). The magnetic catalyst was evaluated for the synthesis of pyrano[2,3-*c*]-pyrazoles (**57**) through the one-step, multicomponent reaction of 3-methyl-pyrazolone (**8**), malononitrile (**3**) and substituted aldehyde (**56**) in green solvent ethanol at RT. The nanocatalyst performed significantly and afforded excellent product yields (90–98%) with less reaction time (30 min) (Scheme 15). Heteroaromatic and alkyl aldehydes had longer reaction times and gave lesser yields, as compared to the aromatic aldehydes. The catalyst was recovered from the reaction mixture using a magnet. The material was reused for five runs without significant activity loss

Niya et al. reported [74] an efficient Fe_3_O_4_@THAM-SO_3_H material as a reusable heterogeneous catalyst for the one-pot synthesis of pyrano[2,3-*c*]-pyrazole derivatives (**59**). The resultant nanocatalyst, Fe_3_O_4_@THAM-SO_3_H, was characterised by spectroscopy systems like FT-IR, P-XRD, TGA, DTA, VSM, FE-SEM, EDS, and TEM analyses. The FESEM and TEM analyses proved that the nanocomposite was spherical and uniform, with good dispersity and average size (14 nm). The multicomponent reaction was carried out between the malononitrile (**3**), hydrazine hydrate (**11**), ethyl acetoacetate (**12**), various aldehydes (**58**) and 10 mg catalyst in an equal amount of aqueous ethanol solvent media (Scheme 16). This approach offered many benefits: simple handling, easy workup, short reaction time (25 min), the use of green, non-toxic solvents, no need for column chromatography, good-to-excellent yields (69–85%), and a catalyst that was recycled eight times with sustained activity.

## 3. With Bi-Metallic Catalysts

Bimetallic substances are well-defined as a combination of two metals, either in composites or attached metal fragments. Subsequently, the construction of bimetallic catalysts is a significant step to take towards improving their properties. These can be as deliberate as catalyst particles designed in nano-sizes ranging 10 to 100 nm. In bimetallic compounds, one element may enhance activity covering the second element’s active sites. The use of bimetallic nanoparticles is quite unusual in that it avoids the use of alloy structures. Various procedures involve catalysts preparations that are eco-friendly and environmentally safe, and properties depend on metal oxide loading and scums. In rare circumstances, one metal can be stabilised in a low valence state to support the active metal. However, in highly distributed bimetallic nanoparticles, one can’t ignore the partial or overall decrease of this interface when applied to diverse systems.

Maddila et al. [75] developed manganese-doped zirconia (Mn/ZrO_2_) which was prepared via the wet impregnation method. The spherical morphology and the crystallite size (12–23 nm) of the catalyst material was established by SEM and TEM investigation. Additionally, N_2_ adsorption-desorption studies showed the catalyst surface area (194.56 m^2^/g) and pore volume (0.563 cc/g). The Mn/ZrO_2_ catalyst was applied for the novel synthesis of two series of pyrano[2,3-*c*]-pyrazoles (**62** and **63**) from a one-pot, multicomponent reaction of ethyl acetoacetate (**12**)/dimethylacetylenedicarboxylate (**61**), malononitrile (**3**), hydrazine hydrate (**11**), aryl aldehydes (**60**) and ethanol (5 mL) solvent under ultrasound irradiation at 80 °C. Excellent product yields (88–98%) were obtained in 10 min reaction time using Mn/ZrO_2_ catalyst (30 mg) (Scheme 17). This reaction sequence involved Knoevenagel condensation, Michael addition and cyclisation sequence. With similar efficiency, the catalyst was reused for up to six cycles. This protocol offered reusability, eco-friendliness, a simple workup and a swift reaction time.

Heravi et al. [76] reported an efficient, multicomponent and straightforward preparation of dihydropyrano[2,3-*c*] pyrazole scaffolds (**65**) using Preyssler-type hetero polyacid (H_14_[NaP_5_W_30_O_110_]) as a recyclable catalyst via a one-pot reaction. A multicomponent reaction employed malononitrile (**3**), a variety of aldehydes (**64**) and 3-methyl-pyrazolone (**8**) at refluxing conditions for 60 min. Additionally, 1 mol% of catalyst effectively performed under ethanol or water media, affording excellent yields (85–94%) of the target molecules (Scheme 18). Authors also examined the various active methylene compounds (such as diethyl malonate, ethyl acetoacetate, ethyl cyanoacetate, ethyl benzoyl acetate and acetophenone) under similar reaction conditions. The protocol was only dynamic with malononitrile. The reaction progressed via Knoevenagel condensation (**66**), Michael addition (**67**) and was followed by intramolecular cyclisation (**69**) to yield the desired product. The catalyst was reused five times with sustainable activity. Excellent yields, reusability, simple workup procedure and green solvents were the primary merits of this protocol.

Pradhan et al. [77] synthesised a highly efficient nanocatalyst, copper ferrite (CuFe_2_O_4_), using the simple citric acid complex method. Attributable to its Lewis acidic sites, the catalyst worked efficiently for the synthesis of pyrano[2,3-*c*]-pyrazoles (**74** and **75**) from the four-component reaction of alkyl nitrile (malononitrile and ethyl cyanoacetate) derivatives (**70**), various hydrazine derivatives (**71**), dialkyl acetylenedicarboxylate (**72** and **73**) and ethyl acetoacetate (**12**) (Scheme 19). Additionally, 8 mol% of CuFe_2_O_4_ gave excellent yields (85–97%) in water at 60 °C for 4 h reaction time. The reaction mechanism proposed that first, a Knoevenagel condensation occurred between hydrazine and ethyl acetoacetate. In the second step, a Michael reaction occurred between malononitrile and diethyl acetylenedicarboxylate, possibly due to Cu^2+^ active sites. The above two intermediates cyclised together via intramolecular action, facilitated by the Fe^3+^ Lewis acidic sites and Cu^2+^ active sites. Unsatisfactory yields (12–43%) of the desired products were obtained by the replacement of the ethyl acetoacetate (**36**) to the dialkyl acetylenedicarboxylate (**46**). This protocol offered several benefits, such as the simple workup, highly functional group tolerance, avoidance of harmful solvents, eco-friendly conditions, and significant yields.

Zolfigol et al. [78] successfully prepared an efficient and stable nano-Fe_3_O_4_@SiO_2_@(CH_2_)3-Imidazole-SO_3_HCl as a magnetic catalyst. Using FT-IR, DTA, TGA, SEM, SEM-EDX, TEM, BET, P-XRD, and ICP analysis, the authors established the prepared catalyst structure. TGA and DTA analyses showed a 2% weight loss and thermal decomposition at 550 °C in three stages. The presence of nanocatalyst-expected elements, C, Cl, Fe, N, O, S and Si, was confirmed by EDX analysis. The nanocatalyst P-XRD pattern exhibited crystallite size (40 nm), in good agreement with TEM and SEM analysis (12.42–57.12 nm). The BET spectrum of nanoparticles showed the specific surface area (129 m^2^ g^−1^), pore volume (0.281 cc g^−1^) and diameter (8.77 Å). The activity of the nanocatalyst was assessed on the three-component reaction of varieties of aldehydes (**76**), malononitrile (**3**), and 3-methyl-pyrazolineone (**8**) to synthesise dihydropyranopyrazole derivatives (**77**). Remarkably, 85–98% of the target yields were obtained under solvent-free conditions with 0.0007 g of catalyst at RT for <90 min (Scheme 20). Both aromatic and heteroaromatic aldehydes, bearing electron-donating or electron-withdrawing groups, were well tolerated and gave the products’ high yields. The nanocatalyst was recovered by simple magnetic separation and reused for six cycles with only a minor loss of activity.

Maddila et al. [79] developed ceria-doped zirconia (CeO_2_/ZrO_2_) as a heterogeneous catalyst and prepared it by the wet impregnation method. The CeO_2_/ZrO_2_ catalyst was performed by the synthesis of novel pyranopyrazoles (**79**) with excellent yields (88–98%) in 15 min at RT (Scheme 21). The four-component reaction—comprising malononitrile (**3**), hydrazine hydrate (**11**), ethyl acetoacetate (**12**) and substituted aldehydes (**78**) in ethanol (5 mL)—was effectively catalysed by 50 mg catalyst. The catalyst was recovered by filtration and recycled for six cycles with sustained efficiency. The protocol offered high atom economy, avoided the use of a column and hazardous chemicals, and boasted a simple workup procedure and swift reaction time.

Fatahpour et al. [80] developed a nano-thin film, Ag/TiO_2,_ as a recyclable catalyst for the preparation of pyrano[2,3-*c*]-pyrazoles (**81**) from the four-component reaction between ethyl acetoacetate (**12**), malononitrile (**3**), structurally substituted aldehydes (**80**) and hydrazine hydrate (**11**). The nanocatalyst performed well with various aldehydes to give 26 dihydropyrano-[2,3-*c*]-pyrazole derivatives with excellent yields (78–93%) at 70 °C for 55 min using aqueous ethanol (Scheme 22). The proposed reaction mechanism involved the Knoevenagel condensation, Michael addition and intramolecular cyclisation via tautomerisation to yield the desired products. Furthermore, the nano-film catalyst was stable and able to be reused ten times with sustained activity.

Uderji et al. [81] reported a new, high-surface-area (509.5 m^2^/g) mesoporous catalyst (Fe_3_O_4_@FSM-16-SO_3_H) for the preparation of novel dihydropyrano-[2,3-*c*]-pyrazole scaffolds (**83**). FESEM study showed that the nanoparticles had a spherical morphology with size < 100 nm. The target molecules were prepared by a multicomponent reaction between malononitrile (**3**), 3-methyl-pyrazolone (**8**) and aldehydes (**82**) in the presence of Fe_3_O_4_@FSM-16-SO_3_H (30 mg) in a water-ethanol system in high yields (79–89%) for 80 min at reflux condition (Scheme 23). The recovered catalyst was reused five times with similar activity.

A facile, three-component reaction of malononitrile (**3**), 3-methyl-pyrazolinone (**8**), and aromatic aldehydes (**84**) leading to pyranopyrazole derivatives (**85**) progressed well under a solvent-free condition in 120 min at 80 °C with the aid of a low catalytic amount (0.003 g) of Fe_3_O_4_@TiO_2_@(CH_2_)_3_OWO_3_H via one-pot manner (Scheme 24) [82]. The prepared novel catalyst was assessed by several techniques, including SEM, EDS, FT-IR, P-XRD, and VSM analysis. SEM analysis specified that the catalyst showed uniform spherical shape particles with an average size (34–91 nm), which was in good agreement with XRD investigation. The EDX study confirmed C, Fe, O, Ti and W in the nanocomposite. Additionally, SEM and VSM showed that the deposit procedure improved particle size and reduced the catalyst material’s magnetic possessions. The MCR proceeded smoothly, with a domino Knoevenagel and Michael addition via cyclisation reaction. The catalyst was stable for five cycles with no marked change in the activity. The merits of this protocol included the simple workup procedure, superior yields (78–92%), swift reaction time and reusability.

A new protocol for the synthesis of pyranopyrazole scaffolds (**87**) with nanomagnetic iron material [CoFe_2_O_4_] as a reusable catalyst in aqueous solvent was described by Mishra and co-workers [83]. The one-pot approach, in the presence of a catalyst (0.05 g), involved malononitrile (**3**), hydrazine hydrate (**11**), ethyl acetoacetate (**12**) and substituted aldehydes (**86**), and used ultrasound irradiation for 5 min (Scheme 25). Both electron-withdrawing and electron-donating groups worked well and gave significant yields of the products. The probable mechanism proceeded with a tandem reaction between hydrazine hydrate and ethyl acetoacetate, resulting in prompt production of an intermediate (pyrazolone) and aromatic aldehyde and an active methylene group to produce Knoevenagel adduct. Further, Michael addition of the intermediate pyrazolone to Knoevenagel adduct, followed by intramolecular cyclisation and rearrangement, yielded the target molecule.

A highly efficient recyclable nanocatalyst, yttrium iron garnet (Y_3_Fe_5_O_12_; YIG), was described by Sedighinia et al. in 2019 [84]. The YIG (0.005 g) catalyst was used for synthesising pyranopyrazoles (**89**), fusing hydrazine hydrate (**11**), malononitrile (**3**), ethyl acetoacetate (**12**) and substituted aldehydes (**88**) under solvent-free conditions at 80 °C with excellent yields (89–95%) in 20 min (Scheme 26). Both electron-donating and electron-withdrawing groups at the ortho, meta and para positions were well-tolerated, giving excellent yields. The easily-wrecycled nanocatalyst was used for eight runs with similar activity.

A highly efficient and eco-friendly magnetic silica-supported propylamine/molybdate composite (Fe_3_O_4_@SiO_2_/Pr-N=Mo[Mo_5_O_18_]) was designed by Neysi et al. [85] for the preparation of novel pyrano[2,3-*c*]-pyrazole analogues (**91**). The catalyst complex’s core shell was assessed using TGA, FT-IR, SEM-EDX, VSM, and P-XRD analysis. The TEM and SEM analysis revealed a spherical particle distribution and core-shell morphology with average size (80 nm) for the prepared catalyst. Furthermore, the EDX study showed C, Fe, Mo, N, O and Si elements, all of which were homogeneously dispersed in the complex. The one-pot reaction of malononitrile (**3**), ethyl acetoacetate (**12**), hydrazine hydrate (**11**), and substituted aldehydes (**90**) was successfully catalysed by core-shell catalyst in water for reaction time (20 min) at RT (Scheme 27). The magnetic catalyst was easily recoverable and stable at 500 °C. The material was reused ten times with a minimal loss of activity.

Mohtasham’s research group developed an efficient H_3_PW_12_O_40_-loaded, aminated magnetic (Fe_3_O_4_@SiO_2_-EP-NH-HPA) nanocatalyst for the novel preparation of pyrano[2,3-*c*]-pyrazole scaffolds (**93**) [86]. The nanocatalyst exhibited high efficiency in the four-component reaction of malononitrile (**3**), hydrazine hydrate (**11**), ethyl acetoacetate (**12**), and various aldehydes (**92**) in a water medium at room temperature, affording excellent yields (89–99%) in 10 min (Scheme 28). The magnetic nanocatalyst was recycled and reused up to seven times without noticeable loss of its catalytic activity. Both electron-donating and electron-withdrawing groups at the ortho, meta and para positions of the aldehydes were well-endured and offered excellent yields.

## 4. With Tri-Metallic Catalysts

The design of nanomaterials, combining various elements with unique characteristics, is a crucial step towards resolving multiple problems and challenges in catalysis—as well as other applications. In recent years, novel and special features of materials combining three distinct metal elements to the alloy have attracted considerable attention in catalytic systems. These materials typically afford improved or exclusive properties due to innumerable synergistic effects. Compared to bimetallic and monometallic compounds, trimetallic materials can be made to exhibit higher degrees of selectivity, efficiency, and catalytic activity by varying their elemental composition and morphologies. The synthesis of trimetallic nanoparticles has been of great interest, due to their multiple potential applications in semiconductors, biosensors and the medicinal and catalysis fields.

In 2013, a swift and highly efficient lanthanum strontium magnesium oxide (La_0.7_Sr_0.3_MnO_3_ (LSMO)) was prepared as a mixed heterogeneous catalyst by Azarifar et al. [87]. Authors characterised the material using various techniques, including SQUID spectral analysis. The catalyst composite exhibited exceptional surface area (39 m^2^/g) and average size (ca.20 nm) and magnetisation (ca 15 emu. gm^−1^) properties. Use of 5 mol% of LSMO catalyst material facilitated the synthesis of novel pyrano-[2,3-*c*]-pyrazole scaffolds (**21**) under ultrasound irradiation conditions from the reaction between malononitrile (**5**), different aromatic aldehydes (**20**) and 3-methyl-pyrazolinone (**7**) in ethanol medium (Scheme 29). Both electron-withdrawing and -donating substituents were well-tolerated, giving excellent yields (86–95%) of the target products at room temperature after 10 min.

Maleki1 et al. [88] demonstrated a series of pyrano[2,3-*c*]-pyrazoles (**99**) and tetrahydrobenzo-pyrans (**100**) using an effective silica-coated magnetic catalyst (NiFe_2_O_4_@SiO_2_–H_3_PW_12_O_40_; NFS–PWA) as a heterogeneous acidic catalyst. FTIR analysis confirmed the presence of NFS-PWA peaks at 807, 888 and 984 cm^−1^, signifiying the effective hold of the PWA on NFS’s surface. Furthermore, TEM and SEM morphology showed that catalyst particles were a regular spherical shape and narrowly dispersed, with sizes ranging from 28 to 97 nm. The nanocatalyst revealed high efficacy in the three-component reaction of 3-methyl-pyrazolineone (**8**), active methylene compound (**96**), and various functional groups substituted aldehydes (**97**) under ethanol solvent conditions. Use of 0.02 g of the nanocatalyst gave superior results (76–94%) within a 10 min reaction time at reflux conditions (Scheme 30). Furthermore, replacement of 1,3-dicarbonyl compound (**98**) to the 3-methyl-pyrazolineone (**8**) resulted in the significant yields (80–95%) of tetrahydrobenzopyrans (**100**) with similar reaction conditions after 15 min. The catalyst material showed matching activity for five runs and offered excellent yields of both the tetrahydrobenzopyran and pyranopyrazole derivatives within a short reaction time.

Azarifar et al. [89] synthesised a novel *γ*-Fe_2_O_3_@Cu_3_Al-LDH-TUD material which showed superior catalytic activity in the synthesis of new dihydropyranopyrazoles (**103**) from the corresponding 3-methylphenylpyrazolinone (**8**), malononitrile (**3**), and substituted aldehyde (**101**) at 100 °C for 45 min under solvent-free conditions. SEM, XRD, EDX and FT-IR spectroscopic techniques characterised the configuration of the *γ*-Fe_2_O_3_@Cu_3_Al-LDH-TUD catalyst. The P-XRD analysis indicated that diffraction peaks at 9°, 30°, 35°, 43°, 57° and 63° agreed with *γ*-Fe_2_O_3_@Cu_3_Al-LDH-TUD catalyst. Further, SEM analysis showed spherical morphology and average particle size (22 nm). Similarly, this protocol proved good with 4-hydroxy coumarin (**102**) for the preparation of dihydropyrano[3,2-*c*]-chromenes (**104**) with significant yields (75–93%) (Scheme 31). The multicomponent reaction was subjected to the steric and electronic influences connected with various substituents on aldehydes. Reportedly, the sequence initiated by malononitrile and aldehyde reacted to generate an intermediate via Knoevenagel condensation. In the follow-up, the Michael addition of 3-methylphenylpyrazolinone to the intermediate delivered the preferred target molecule.

## 5. With Miscellaneous Materials and Composites as Catalysts

Guo et al. [90] developed a series of pyranopyrazoles (**108**) and spiro[indoline-pyrano[2,3-*c*]-pyrazole] derivatives (**109**) using meglumine as a biodegradable catalyst. The four-component reaction, comprising malononitrile (**3**), hydrazine hydrate (**11**), β-keto ester (**12**) and carbonyl compound (**106**) or isatin (**107**) in EtOH-H_2_O solvent in the presence of 10 mol% meglumine at RT, gave excellent yields (85–95%) (Scheme 32). The catalyst was reusable for up to 3 cycles with trivial loss of activity. A reaction mechanism involving the Knoevenagel condensation (**114**), enolate formation (**112**) in the presence of meglumine from ethyl acetoacetate and hydrazine, Michael type addition (**115**), intramolecular cyclisation (**116**) and tautomerisation (**117**) was suggested, in order to deliver the desired pyranopyrazoles.

Bora and his research group [91] described the preparation of dihydropyrano-[2,3-*c*]-pyrazoles (**120** and **121**) in the presence of *Aspergillus niger* Lipase (ANL) catalyst. Lipase effectively catalysed the four-component condensation of hydrazine hydrate (**11**), malononitrile (**3**), ethyl acetoacetate (**12**) and aldehyde (**118**) or ketone (**119**) in ethanol medium. Use of 20 mg of ANL catalyst resulted in excellent yields (70–98%) at 30 °C for 1 h reaction time (Scheme 33). The biocatalyst was reused 3 times without loss of its catalytic efficiency. Some of the main advantages of this approach were its broad substrate scope, easy workup, excellent yields, biocatalysts, green solvent and eco-friendliness.

Xingtian and his co-workers [92] effectively synthesised bovine serum albumin (BSA) and used it as a catalyst for the generation of new pyrano[2,3-*c*]-pyrazole derivatives (**123**). The catalytic performance of BSA was assessed by the four-component reaction between hydrazine hydrate (**11**), malononitrile (**3**), ethyl acetoacetate (**12**) and a wide range of carbonyl compounds (**122**) in an ethanol system at 45 °C for 45 min reaction time, obtaining excellent yields (84–95%) (Scheme 34). Initially, the amino-functional group on basic BSA concurrently helped the reaction between hydrazine hydrate and ethyl acetoacetate to form olefin via condensation. The reaction of an aromatic aldehyde with malononitrile afforded 3-methyl-1*H*-pyrazol-5(4H)-one as other intermediates via Knoevenagel condensation. Further, Michael addition of 3-methyl-1*H*-pyrazol-5(4*H*)-one and olefin facilitated by BSA generated the intermediate. Lastly, the new intermediate was cyclised via intramolecular cyclisation, followed by tautomerisation to afford the target compounds. Recycling investigations showed that the recovered BSA could be reused for up to five runs with similar efficiency.

In 2017, Nazari and his colleagues [93] developed a novel amberlite-supported L-prolinate ([Amb]L-prolinate). The material showed good catalytic efficacy in synthesising pyrano-[2,3-*c*]-pyrazole scaffolds (**125**) through the one-pot approach. The multicomponent condensation reaction, involving malononitrile (**3**), 3-methyl-pyrazolone (**8**), and varied substituted aldehydes (**124**) under ethanol medium at reflux conditions, afforded excellent yields (85–98%) of the target molecules within a short reaction time (<25 min), employing 10 mol% [Amb]L-prolinate as the catalyst (Scheme 35). The one-pot reaction occurred through the Knoevenagel condensation and Michael addition via cyclisation in sequence. The catalyst was usable for up to eight cycles with sustained efficiency.

A novel and efficient zwitter-ionic sulfamic acid-functionalised clay (MMT-ZSA) nanocatalyst was synthesised by Safari and co-workers via the silane condensation process [94]. The MMT-ZSA nanocatalyst was used in the preparation of dihydropyrano-[2,3-*c*]-pyrazoles (**128**) via a one-pot reaction of malononitrile (**3**), β-keto ester (**12**), hydrazine derivatives (**127**), and carbonyl compounds (**126**). The reaction in the absence of solvent at 90 °C for 15 min time afforded excellent (83–95%) yields (Scheme 36). All the aldehydes, irrespective of electron-withdrawing or electron-donating groups, afforded good yield in the four-component reactions.

Devi and her research group [95] reported an efficient one-pot synthesis of spiro-indoline-pyranopyrazoles (**131**) in the presence of sodium dodecyl sulfate (SDS) as a micelle catalyst. The three-component condensation was carried out between the isatin (**129**), malononitrile (**3**) and 3-methyl-pyrazol-one (**130**) under an aqueous medium. A 2.5 mol% of SDS catalyst offered excellent (80–91%) yields at RT for 60 min (Scheme 37). The micelle catalyst was recyclable with similar activity up to four runs. The stated reaction mechanism giving the desired spiro products was involved in the Knoevenagel condensation (**132**) and Michael addition (**133**), and was followed by cyclisation (**134**).

Farokhian and co-workers [96] developed an efficient and novel sulfonic acid-functionalised ionic liquid [DMBSI]HSO_4_ catalyst for the one-pot synthesis of pyrano[2,3-*c*]-pyrazoles with higher yields (85–95%) (**137**) in a short reaction time (20 min) at 90 °C (Scheme 38). The four-component condensation reaction was implemented with dimethyl malonate (**136**), ethyl acetoacetate (**12**), aromatic aldehyde (**135**), hydrazine hydrate (**11**) and [DMBSI]HSO_4_ (0.18 mmol) under solvent-free conditions. The plausible mechanism then proceeded to the resulting tandem reactions—Knoevenagel condensation, Michael addition and cyclisation in sequence—to obtain the target molecules.

Shinde and his research group [97] designed the successful synthesis of a beal fruit ash (BFA) as a natural catalyst. The resultant BFA catalyst was confirmed and characterised by various spectroscopy analysis, i.e., FT-IR, XRD, SEM and DSC-TGA. The efficacy of the BFA was evaluated in the synthesis of pyrano[2,3-*c*]-pyrazoles (**140**) and pyrazolyl-4*H*-chromene (**141**) derivatives via the four-component reactions of malononitrile (**3**), hydrazine hydrate (**11**), ethyl acetoacetate (**12**), various aldehydes (**138**) and water as a medium, at room temperature for 30 min (Scheme 39). The procedure proved ideal for the generation of pyrazolyl-4*H*-chromenes (**141**), with excellent yields (86–94%) from different salicylaldehydes (**139**) after 15 min. The catalyst was stable for up to 5 runs with little loss in its activity.

Mejdoubi et al. [98] described an efficient, recyclable catalyst, natural phosphate K09, which was used for synthesising pyrano[2,3-*c*]-pyrazoles (**133**). The EDX study revealed the catalyst material’s elemental composition as Al, Cd, Cr, Fe, K, Na, S, Si, and Mg. The BET analysis of the K09 catalyst showed the specific surface as 13.8 m^2^.g^−1^. The one-pot reaction between malononitrile (**5**), hydrazine hydrate (**22**), ethyl acetoacetate (**36**) and aryl aldehydes (**132**) in ethanol medium gave excellent results (90–98% yield) using 0.05 g of K09 catalyst at RT in 20 min (Scheme 40). The catalyst was stable up to six times, with a minute change in its efficiency. Both aromatic aldehydes and heteroaromatic aldehydes contributed excellent yields.

An environmentally-benign, sulfonated carboxymethyl cellulose (CMC-SO_3_H) catalyst was synthesised by Ali et al. for the creation of novel pyrano[2,3-*c*]-pyrazole derivatives with good-to-excellent yields (78–90%) (**145**) [99]. The catalyst was investigated by a multicomponent fusion between ethyl acetoacetate (**12**), malononitrile (**3**), hydrazine hydrate (**11**) and various substituted aldehydes (**144**) in ethanol at 60 °C via a one-pot approach (Scheme 41). The reaction mechanism included Knoevenagel condensation, Michael-type addition and intramolecular cyclisation to obtain the resultant target molecules. The heterogeneous catalyst was fully stable for up to four runs with marginal loss of activity.

An efficient and novel magnetically-separable catalyst, Cu^2+^ doped Ni-Zn nano ferrite composite (Ni_0.5_Cu_0.3_Zn_0.2_Fe_2_O_4_), was designed by Mandle et al. via the sol-gel approach [100]. The four-component reaction of pyranopyrazoles (**147**) from malononitrile (**3**), hydrazine hydrate (**11**), ethyl acetoacetate (**12**), and various substituted benzaldehydes (**146**) was catalysed by 20 mg of Ni_0.5_Cu_0.3_Zn_0.2_Fe_2_O_4_ catalyst in the presence of ethanol under reflux condition (Scheme 42), giving excellent yields (91–95%) in 30 min. The effortlessly-separable catalyst was reused five times with marginal loss of efficiency.

Ganesan and his co-workers [101] reported a swift synthesis of pyranopyrazole derivatives (**149**) by applying nitrogen-doped graphene oxide (NGO) as a recyclable catalyst. The prepared NGO catalyst’s SEM analysis showed unsystematically-aggregated, narrowly-connected tinny sheets with various morphologies. Further, TEM displayed a well-ordered and distinct sheet and a silk-like morphology. Additionally, HR-XPS analysis strongly demonstrated the presence of multiple types of nitrogens in the NGO catalyst. The condensation reaction involved malononitrile (**3**), ethyl acetoacetate (**12**), hydrazine hydrate (**11**), and various functional groups of substituted aldehydes (**148**) under solvent-free conditions. Excellent results (80–99%) were obtained using 10 mg of NGO catalyst in 2 min through the grinding approach (Scheme 43). It is noteworthy that the stable catalyst material could be recycled and reused up to eight successive cycles with only a slight efficiency decrease.

## 6. Conclusions

This review outlined the increased available selection of recyclable catalyst materials, which are ideal for organic synthesis in general and pyranopyrazoles in particular and are expanding the scope and utility of heterogeneous catalysts. Simultaneously, the assessment of the implementation of multicomponent reactions under green techniques offers good compatibility with sustainable organic syntheses. With many green chemistry principles, the intrinsic features of the MCR approach becomes the foremost device in the synthetic chemist’s toolbox. Nitrogen-based pyranopyrazole compounds are constituents of several natural products with numerous pharmaceutical applications. Thus, these emerging innovative and versatile approaches for synthesising these heterocyclic skeletons have always been challenging and rewarding. This review compiled recent literature, detailing the scope and broader choice of heterogeneous metal-based catalysts available to create the prized fused heterocycles, pyranopyrazoles. In this perspective, we emphasised the efficiency of MCRs as the ideal process for green synthesis. In most of the protocols, the employed catalysts were easily recyclable for successive runs with consistent performance. We sincerely hope this article will help synthetic chemists to develop novel heterogeneous materials and synthetic routes with greater efficiency and sustainability.

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
