# Peer review of "Advances in Pyranopyrazole Scaffolds’ Syntheses Using Sustainable Catalysts—A Review"

_molecules, 2021, doi:10.3390/molecules26113270_

Round 1

Reviewer 1 Report

The review paper by Jonnalagadda and co-workers describes extensively the merits of heterogeneous catalyst use in MCR-approaches to access pyranopyrrazole scaffolds. These are highly valued heterocyclic core fragments that are utilized especially in medicinal chemistry. The paper is comprehensive and addresses all the major points in this field. As such a valuable contribution and I can support pubication in Molecules after the authors have addressed some minor issues pointed out below:

  • Especially the introduction is written in an "over the top" style. I understand that authors want to "sell" their work well, but this is too much. I suggest rewriting the complete introduction in a more down to earth style. Use much shorter sentences, that will make the text much more readable. Also I think the introduction would benefit from a more structured analysis using perhaps sub-headings. Moreover one or two more conceptual pictures/schemes would also help understanding the major points that will be addressed in this review paper.
  • A second point of concern that I noticed is that in the references section not always the correct citation is used. The authors mix-up first and last names of authors in their references, like e.g. in reference 13. This may occur at other places as well. please do carefully check this throughout the manuscript.

Author Response

Pointwise responses to reviewer are attached

Reviewer 2 Report

The manuscript received for evaluation, titled 'Advances in pyranopyrazole...' by Jonnalagadda et al. has as main topic a review about the mentioned compounds in organic chemistry and similar areas, as catalysts and so on, namely in the synthesis of nitrogen-based fused heterocyclic ring frameworks, using diverse recyclable catalysts, etc.

The timespan cover is the last 10 years. The subject chosen can be of high interest for researchers, working in chemistry and materials areas, but not excluding bio applications. It is highlighted that heterocycles are imperative components of many natural materials and such compounds are extremely valuable in organic and medicinal chemistry. The review is comprehensive and meet all the requirements for publication. Thus, it has over 40 Schemes and over 100 references, written in the correct format.

Besides, this review outlines the increased choice of recyclable catalyst materials to be used for synthetic purposes for the obtaining of pyranopyrazoles. In this way the utility of heterogeneous catalysts is evidenced. Moreover, multicomponent reactions employing green techniques lead to sustainable ways in organic syntheses, that can be considered as green chemistry. It is well known that pyranopyrazole compounds are constituents of many natural products and so they have low toxicity and a lot of possible pharmaceutical applications.

There are no weak points regarding the style or the science, therefore the recommendation is to be accepted for publication as it is.

Author Response

Response to reviewer's comment is attached.
